# Transcriptome and Endogenous Hormone Analysis Reveals the Molecular Mechanism of Callus Hyperhydricity in Flax (*Linum usitatissimum* L.)

**DOI:** 10.3390/ijms26115360

**Published:** 2025-06-03

**Authors:** Dandan Liu, Lili Cheng, Lili Tang, Lie Yang, Zhongjuan Jiang, Xixia Song, Qinghua Kang, Dandan Yao, Si Chen, Jiarong Ru, Lili Zhang, Guangwen Wu, Hongmei Yuan

**Affiliations:** 1Heilongjiang Academy of Agricultural Sciences, Harbin 150086, China; ldd5020@163.com (D.L.);; 2College of Advanced Agriculture and Ecological Environment, Heilongjiang University, Harbin 150086, China

**Keywords:** hyperhydricity, callus, transcriptome, hormone, *Linum usitatissimum* L.

## Abstract

Hyperhydricity is a frequently occurring physiological disorder in plant tissue culture that impairs the regeneration and survival of vitrified plant materials, leading to significant losses and complicating research applications. Although numerous studies have focused on strategies to mitigate hyperhydricity, its molecular mechanisms remain poorly characterized. In this study, transcriptome sequencing and endogenous hormone content analysis were conducted on hyperhydric and normal callus of flax (*Linum usitatissimum* L.). Transcriptomic analysis revealed 2698 differentially expressed genes (DEGs) between these two tissue types. Pathway analysis through KEGG demonstrated that these DEGs predominantly linked to metabolic processes including phenylpropanoid formation, MAPK signaling cascades, and plant hormone signal transduction. Moreover, quantification of endogenous hormone levels indicated a significant reduction in all hormones except cytokinins (ZRs) in hyperhydric tissues. The observed disruption in endogenous hormone levels suggests its potential role in the development of plant vitrification. These findings provide valuable insights into the molecular processes underlying hyperhydricity, contributing to a more comprehensive understanding of this phenomenon.

## 1. Introduction

Plant tissue culture technology is an extensively utilized and powerful technique that has been used in virus-free seedling production, plant breeding enhancement, and rapid propagation. However, a significant challenge associated with in vitro culture is hyperhydricity (also referred to as vitrification or glassiness), a physiological disorder that negatively affects plant growth and development [1]. This condition results from excessive water uptake, leading to various morphological and anatomical anomalies, including translucent, thick, fragile, and curled leaves [2,3]; poorly developed epidermis and cuticle; malformed stomata; an accumulation of starch granules within plastids; a reduction in chloroplast numbers; and enlarged intercellular spaces within mesophyll cells [4,5]. These abnormalities severely impair plant regeneration and survival, thereby limiting the efficacy of plant tissue culture applications.

Hyperhydricity is primarily induced by unfavorable tissue culture conditions such as excessive humidity, suboptimal light exposure, the accumulation of harmful gases, and fluctuations in osmotic pressure [1,6]. Various approaches have been proposed to prevent hyperhydricity, including optimizing light conditions [7], improving the air permeability of culture vessels [8,9], adjusting cytokinin type and concentration [10,11,12], and incorporating specific biochemical agents such as reactive oxygen species scavengers (e.g., salicylic acid) [13,14] or ethylene activity inhibitors (e.g., AgNO_3_) [15,16,17]. Despite these advances, the fundamental molecular and genetic pathways responsible for hyperhydricity are still not well elucidated.

Previous studies have provided molecular insights into hyperhydricity through transcriptomic and proteomic analyses. Bakir et al. identified over 300 differentially expressed transcripts in hyperhydric versus normal peach (*Prunus persica* L.) leaves, with the top 30 related to post-transcriptional regulation, photosynthesis, cell elimination, cuticle development, and abiotic stress response [18]. Similarly, proteomic studies of hyperhydric and normal carnation shoots identified 40 differentially abundant proteins associated with photosynthesis, RNA processing, and metabolism [19]. DNA methylation studies have further indicated that hyperhydric plants show lower methylation levels [20,21]. Furthermore, Lelis et al. demonstrated 19 genes related to hyperhydricity and found that 17 were upregulated in hyperhydric embryos, predominantly associated with stress response, energy metabolism, and defense mechanisms [22].

Flax tissue culture regeneration systems serve as the foundation for genetic enhancement via *Agrobacterium*-mediated genetic engineering. However, hyperhydricity poses a significant barrier to efficient in vitro regeneration and transformation, leading to the inefficient use of research materials and reduced experimental success. Specifically, the molecular mechanisms driving hyperhydricity in flax remain unexplored.

To address this issue, transcriptomic analyses were conducted on hyperhydric and non-hyperhydric flax callus tissues, and an assessment of endogenous hormone content was undertaken to elucidate the molecular mechanisms contributing to hyperhydricity. The outcomes of this investigation may establish a crucial basis for the prevention and management of hyperhydricity in flax tissue culture.

## 2. Results

### 2.1. Morphologic Observation

Callus tissue with normal physiological characteristics appears bright green, maintains a compact structure, and can sprout normally (Figure 1A). On the other hand, hyperhydric callus tissue is distinguished by its translucent and watery appearance, fragile and friable consistency, reduced potential for cellular differentiation, and challenges in the initiation and development of buds (Figure 1B).

### 2.2. Overview of Transcriptome Sequencing Results

To explore the molecular response to hyperhydricity, a transcriptome-wide analysis was performed on flax callus tissues under normal and hyperhydric conditions. The sequencing process generated 275.14 million raw reads from six specimens (Table 1). Following elimination of low-quality reads, adapter contaminants, and reads containing excessive unknown bases (N), 254.19 M high-quality reads were obtained for subsequent processing. On average, each sample generated 6.36 Gb of sequencing data. Quality assessment revealed that Q20 and Q30 values exceeded 97% and 92%, respectively, confirming the precision and dependability of the sequencing output (Table 1). The mean alignment rate with the reference genome was 96.47%, reflecting a well-assembled and high-quality transcriptomic dataset.

### 2.3. Identification of Differentially Expressed Genes (DEGs)

This study identified DEGs using normal callus (M20) as the control. DEG identification utilized threshold parameters of Log_2_|FC| ≥ 1 and Q-value ≤ 0.05. Sstatistical summaries of DEG identification are presented in Table 2 and Figure 2. A total of 2698 DEGs were detected, comprising 1667 upregulated genes and 1031 downregulated genes. A comprehensive list of these genes can be found in Appendix A. Moreover, the expression profiles of these genes demonstrated significant differences between hyperhydric and control tissues (Figure 3). Clustering analysis clearly distinguished between hyperhydric and control samples, as indicated by their respective groupings.

### 2.4. Functional Annotation of Differentially Expressed Genes

To investigate the functional significance of DEGs, a Gene Ontology (GO) analysis was conducted. The analysis mapped 2698 DEGs across 2927 GO terms, categorized into three primary functional groups: molecular function, cellular component, and biological process (Figure 4A, Appendix A). In the molecular function classification, 69 GO terms exhibited significant enrichment, encompassing DNA-binding transcription factor activity, transcription regulator activity, and heme binding (Figure 4B). The cellular component classification revealed 20 significantly enriched GO terms, including extracellular region, external encapsulating structure, and cell wall (Figure 4B). Additionally, 136 biological process GO terms demonstrated significant enrichment, comprising carbohydrate metabolic process, cellular carbohydrate metabolic process, and cell wall organization (Figure 4B).

To identify the metabolic pathways linked to DEGs, KEGG pathway analysis was executed. A total of 2698 DEGs were assigned to 126 metabolic pathways, which were categorized into five major groups: cellular processes, environmental information processing, genetic information processing, metabolism, and organismal systems (Figure 5A, Appendix A). Using a statistical threshold of Q ≤ 0.05, 19 metabolic pathways were identified as significantly enriched. These pathways primarily involved phenylpropanoid biosynthesis, MAPK signaling, and plant hormone signal transduction (Figure 5B).

### 2.5. Quantitative Real-Time PCR Analysis

To assess the precision of the transcriptome data, a subset of 13 genes potentially associated with hyperhydricity was selected for qRT-PCR validation (Figure 6). The comparative analysis revealed that 12 of these genes (*SOD*, *peroxidase*, *GST*, *USP1*, *USP2*, *SAM*, *ACS1*, *ACO*, *ETR1*, *ETR2*, *ERF14*, and *ERF113*) demonstrated significant upregulation in hyperhydric callus compared to the normal callus. However, *PIP2* showed a significant downregulation. These qRT-PCR findings revealed an overall consistency with the RNA-seq data, thereby validating the accuracy and dependability of the transcriptomic data analysis.

### 2.6. Determination of the Endogenous Hormone Content

Endogenous plant hormones are key regulators of growth and development. In this investigation, functional annotation of DEGs demonstrated significant changes in genes associated with auxin, cytokinin, gibberellin (GA), abscisic acid (ABA), jasmonic acid (JA), and salicylic acid (SA) pathways. To evaluate the physiological impact of these genetic changes, endogenous hormone levels were measured. As depicted in Figure 7, hyperhydric callus tissue showed a significant reduction in most endogenous hormones, except cytokinins (ZRs), which showed no substantial reduction. Indole-3-acetic acid (IAA) levels in non-hyperhydric callus (M20) were significantly higher than those in hyperhydric callus (V20), decreasing from 7.02 ng/mL to 3.58 ng/mL. Similarly, ABA concentrations reduced from 0.49 ng/mL in normal callus to 0.1 ng/mL in hyperhydric callus. The most significant change was observed in SA levels, which dropped from 109.99 ng/mL to 20.91 ng/mL in hyperhydric callus. Furthermore, JA concentrations in M20 were significantly greater than those in V20, decreasing from 2.74 ng/mL to 1.91 ng/mL. Further, GA3 was reduced from 1.42 ng/mL in M20 to 0.45 ng/mL in V20. Cytokinins (ZRs) also decreased, but the change was not significant.

## 3. Discussion

Hyperhydricity represents a frequent physiological disorder in plant in vitro culture, significantly changing the morphology and physiology of plants. This phenomenon presents a substantial obstacle to applying tissue culture in research and contributes to considerable plant material losses. Therefore, elucidating the molecular mechanisms underlying hyperhydricity is crucial to control it.

Hyperhydric callus tissue is characterized by its translucent and excessively hydrated appearance, along with a loose and friable texture (Figure 1). These characteristics resemble those observed in vitrified plants previously studied [23]. Transcriptomic analysis indicated the involvement of various genes in this physiological disorder, with a more significant proportion of genes showing upregulation rather than downregulation (Table 2), consistent with findings reported by Gao et al. [20]. The KEGG pathway examination indicated that numerous DETs displayed enrichment in signal transduction and phenylpropanoid biosynthesis pathways.

Hyperhydric tissue demonstrates numerous abnormalities, many of which may be associated with oxidative stress-induced damage [19,24]. Excessive water uptake results in the overproduction of reactive oxygen species (ROS), which detrimentally affect membrane lipids, nucleic acids, enzymatic functions, and other cellular components. To counteract oxidative stress, hyperhydric tissues show an upregulation of antioxidant enzymes, encompassing catalase, superoxide dismutase, peroxidase, and glutathione S-transferase [4,18,25,26]. The present study demonstrated an increased expression of *superoxide dismutase*, *peroxidase*, and *glutathione S-transferase* in the V20 treatment compared to the control (M20) (Figure 6, Appendix A). Furthermore, two types of universal stress proteins (USPs), Lus10029850 and Lus10025033, were identified, which are known to enhance plant tolerance to abiotic stress [27,28,29].

Previous studies have indicated that excessive ethylene accumulation is a critical factor in developing hyperhydricity [9,20]. Under hyperhydric conditions, ethylene promotes ROS production, which, in turn, exacerbates ethylene metabolism [1]. Consistent with previous studies, genes associated with ethylene biosynthesis and signal transduction were found to be upregulated in hyperhydric plants [20,22]. In the current study, expression levels of ethylene biosynthesis-related genes, including *SAM*, *ACS1*, and *ACO*, were elevated in hyperhydric plants (Figure 6, Appendix A). Moreover, the ethylene receptor genes *ETR1* and *ETR2,* as well as the ethylene response factor genes *ERF14* and *ERF113*, were also increased (Figure 6, Appendix A).

An imbalance in endogenous hormones potentially contributes to hyperhydricity. Kataeva et al. reported that cytokinin levels were elevated in hyperhydric tissues compared to normal tissues [30]. However, in this study, cytokinin levels decreased in hyperhydric tissues, though not significantly, which may be attributed to differences in experimental conditions. Hassannejad and Liu demonstrated that salicylic acid supplementation alleviates hyperhydricity [13,14]. Consistently, salicylic acid levels were notably elevated in normally growing callus versus hyperhydric callus in the present study. Furthermore, GA_3_ levels were substantially lower in hyperhydric tissues (0.45 ng/mL) than in normal tissues (1.42 ng/mL), supporting the findings of Jain et al. that increased GA levels promote growth and mitigate hyperhydricity in cultures [31]. Other endogenous hormones, including IAA, ABA, and JA, were also significantly reduced in hyperhydric tissues, potentially disrupting lignin, protein, and nucleic acid biosynthesis, thereby contributing to hyperhydricity. The results of transcriptome analysis were consistent with the hormone content measurement results. The expression levels of key enzymes *EPS1* (Lus10039256), *PBS3* (Lus10003598), and *ICS* (Lus10016992) involved in the SA synthesis process were downregulated in hyperhydric tissues (Appendix A). The expression level of *KAOx* (Lus10033105), a key enzyme in the GA_3_ synthesis process, was also downregulated in hyperhydric tissues (Appendix A). NCED9 (Lus10026185) is a rate-limiting enzyme involved in the biosynthesis of abscisic acid in plants, and its expression was also downregulated in hyperhydric tissues (Appendix A).

Excessive water accumulation in plant tissues is the most prominent symptom of hyperhydricity. Aquaporins facilitate water absorption and transport and are hypothesized to be present at higher concentrations in hyperhydric plants [32,33]. A previous study by Gao et al. suggested that ethylene-regulated phosphorylation of aquaporins enhances water retention in hyperhydric tissues [20]. Similarly, Lelis et al. observed an upregulation of the *PIP2 aquaporin* gene in hyperhydric embryos [22]. However, the present study observed downregulation of *PIP2* gene expression in hyperhydric tissues versus the control (Figure 6, Appendix A), consistent with the study by Bakir et al. [18]. The reasons for this situation are diverse. One possibility is that the regulatory mechanisms of *aquaporin PIP2* in different species (such as woody vs. herbaceous) are different. Another possibility is that the timing of sample collection may play a critical role in elucidating aquaporin function in hyperhydricity. It is possible that aquaporins initially facilitate water influx to induce an abnormal state, but subsequently downregulate their activity to sustain hyperhydricity. Another possibility is that the increase in ethylene content in hyperhydric tissues enhances the phosphorylation level of aquaporin PIP2, greatly increasing the activity of water channels and negatively regulating the expression of the *PIP2* gene. Next, we can establish samples representing various degrees of hyperhydricity (e.g., normal, mild, and severe), analyze the dynamic correlation between the expression of *aquaporin PIP2* and the hyperhydricity process, and simultaneously detect the ethylene content and the expression of key genes within the signaling pathway. We can also overexpress or knock out the *PIP2* gene to observe any changes in the hyperhydricity rate and verify the causal relationship between the downregulation of its expression and the resulting phenotype.

One of the defining characteristics of hyperhydricity is the reduction in lignin content, which manifests as reduced lignification within the vascular system. This phenomenon may be attributed to decreased enzymatic activity associated with the biosynthesis of lignin precursors and their polymerization [1,34]. A study by Caparrós-Ruiz et al. demonstrated that *ZmLac3* accumulates specifically in lignified regions of maize and suggested that this laccase may play an essential role in lignin polymerization [35]. Similarly, research on *Arabidopsis thaliana* has identified *LAC4* and *LAC17* as critical for lignin polymerization, with their double knockout genes resulting in a significant decrease in lignin content [36]. The transcriptome analysis performed in this study revealed that the expression of the *laccase* (Lus10013800) was downregulated relative to the control (Appendix A), consistent with the findings of Bakir et al. [18].

## 4. Materials and Methods

### 4.1. Plant Materials

In this study, callus was induced from flax hypocotyls. Plump flax seeds underwent a sequential sterilization protocol for disinfection. Seeds were initially washed twice with sterile water, followed by immersion in 75% ethanol for 30 s, after which they were rinsed twice with sterile water. A subsequent disinfection step involved treatment with 20% NaClO for 21 min while shaking to support uniform sterilization. Following sterilization, seeds were rinsed four additional times with sterile water before being transferred to an MS solid medium. The culture conditions consisted of an initial four-day incubation at 25 °C in darkness, followed by two days under light exposure. When the hypocotyls had elongated to 6–8 cm, they were cut into 1 cm segments and transferred onto callus induction medium (MS medium supplemented with 0.02 mg/L NAA and 1 mg/L 6-BA). During the callus induction period, these explants were maintained in a growth room at a temperature of 23 ± 2 °C, with a light intensity of 2000 lx and a photoperiod of 16 h per day. The callus induction process lasted 20 days, after which normal and hyperhydric callus were collected for subsequent experiments.

### 4.2. RNA Extraction, cDNA Library Preparation, and Sequencing

RNA isolation occurred from hyperhydric and non-hyperhydric flax tissues through CTAB (Bio Basic, Toronto, Canada) reagent application per the supplier’s protocols. Fragment Analyzer instrumentation determined RNA integrity and concentration. The RNA underwent thermal denaturation to eliminate secondary structures, followed by mRNA isolation via oligo(dT) magnetic bead separation. Fragmentation reagents cleaved the purified mRNA under specific temperature conditions. Random hexamer primers facilitated initial cDNA strand synthesis, succeeded by second-strand generation. End modification procedures were implemented on the synthesized cDNA, incorporating adenine nucleotides at 3′ positions. The subsequent process involved adapter connection utilizing a specialized ligation system. The final step encompassed quality validation of the amplified cDNA libraries and their processing through Illumina sequencing technology.

### 4.3. Sequencing Data Analysis

To ensure data quality, raw sequencing reads were preprocessed using SOAPnuke (v1.4.0), eliminating low-quality sequences, adapter contamination, and reads containing excessive undetermined bases (N). The resulting high-quality reads were mapped to the reference genome (Lusitatissimum_200_v1.0 Phytozome) using HISAT (v2.1.0), followed by alignment to the reference gene sequences using Bowtie2 (v2.2.5). Gene expression levels for each sample were subsequently quantified using RSEM (v1.2.8). Differential gene expression analysis within experimental groups was performed using DESeq, applying a stringent filtering criterion of fold change ≥ 2 and Q-value ≤ 0.05. Expression profiles of DEGs were visualized using pheatmap-generated heatmaps. Functional classification was performed based on GO and KEGG annotation databases, and pathway enrichment analysis was carried out using the phyper function in R. A significance threshold of Q-value ≤ 0.05 was used to identify genes showing statistically significant enrichment.

### 4.4. qRT-PCR Validation

A total of 1 μg of extracted RNA was reverse-transcribed into cDNA using the PrimeScript™ RT reagent Kit with gDNA Eraser (TaKaRa, San Jose, CA, USA). A LightCycler 480 Instrument (Roche Applied Science, Indianapolis, IN, USA) was used to perform qRT-PCR with TB Green Premix Ex Taq II (TaKaRa). GAPDH and EF1A genes were utilized as internal reference genes [37]. Results were analyzed using the 2^−ΔΔCt^ method. Primers for qRT-PCR (Appendix A) were designed using Primer 5.0.

### 4.5. Measurement of the Endogenous Hormone Content

The samples preserved at ultra-low temperatures were retrieved and ground into a dry powder under liquid nitrogen conditions. An appropriate quantity of fresh plant samples was accurately weighed and transferred into a glass test tube. Then, a mixed extraction solution of isopropanol-water-hydrochloric acid was added to the glass test tube, along with 8 µL of an internal standard solution at a concentration of 1 µg/mL. The mixture was shaken at a low temperature for 30 min. Subsequently, dichloromethane was introduced into the mixture, which was then subjected to shaking at a low temperature for an additional 30 min. After shaking, the mixture was centrifuged at 13,000 r/min at a low temperature for 5 min, and the lower organic phase was taken. Under light-protected conditions, the organic phase was dried with nitrogen gas and then redissolved in methanol (0.1% formic acid). The solution was centrifuged at 4 °C for 10 min (13,000× *g*), and the supernatant was passed through a 0.22 µm filter membrane. The levels of IAA, ABA, ZR, SA, JA, and GA3 were determined using an Agilent 1290 high-performance liquid chromatography system coupled with an AB Sciex QTRAP 6500+ mass spectrometer. The detection limit of plant hormones was 0.1 ng/mL.

## 5. Conclusions

This study represents the first comprehensive analysis of the molecular mechanisms underlying flax hyperhydricity. As a complex physiological disorder, hyperhydricity involves the dysregulation of multiple genes. RNA-seq analysis revealed differential expression in 2698 transcripts, with 1667 genes upregulated and 1031 genes downregulated. Moreover, an imbalance in endogenous hormone levels was identified as a potential primary factor contributing to this condition. These findings provide a scientific foundation for developing effective strategies to prevent and control hyperhydricity in flax.

## Figures and Tables

**Figure 1 ijms-26-05360-f001:**
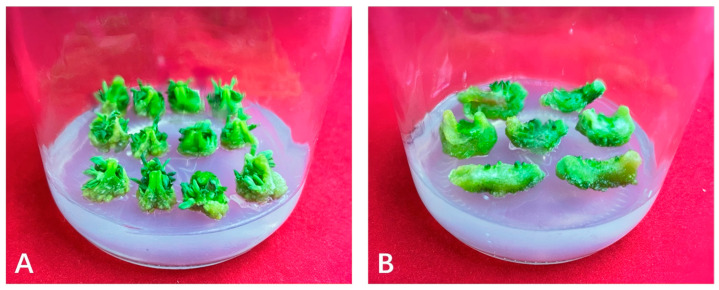
Growth conditions of normal callus and hyperhydric callus. (**A**) Normal callus (M20). (**B**) Hyperhydric callus (V20).

**Figure 2 ijms-26-05360-f002:**
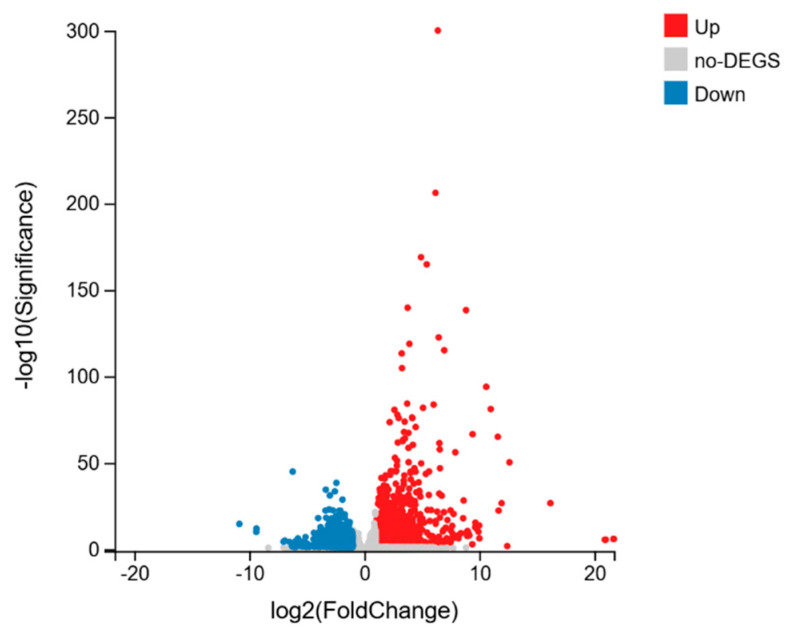
Volcano plot of differentially expressed genes (DEGs). The *X*-axis shows log_2_-transformed fold change values, while the *Y*-axis demonstrates significance values after log_10_ transformation. The plot highlights upregulated DEGs marked in red, downregulated DEGs denoted in blue, and non-DEGs illustrated as gray dots.

**Figure 3 ijms-26-05360-f003:**
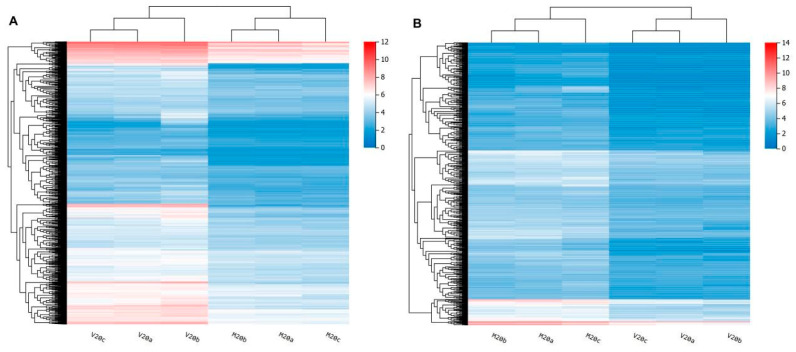
Clustered heatmap of DEGs. (**A**) Clustered heatmap of upregulated genes. (**B**) Clustered heatmap of downregulated genes. The *X*-axis represents the log_2_(FPKM + 1) of the samples, and the *Y*-axis represents the genes. The redder the color of the blocks, the higher the expression level, and the bluer the color, the lower the expression level.

**Figure 4 ijms-26-05360-f004:**
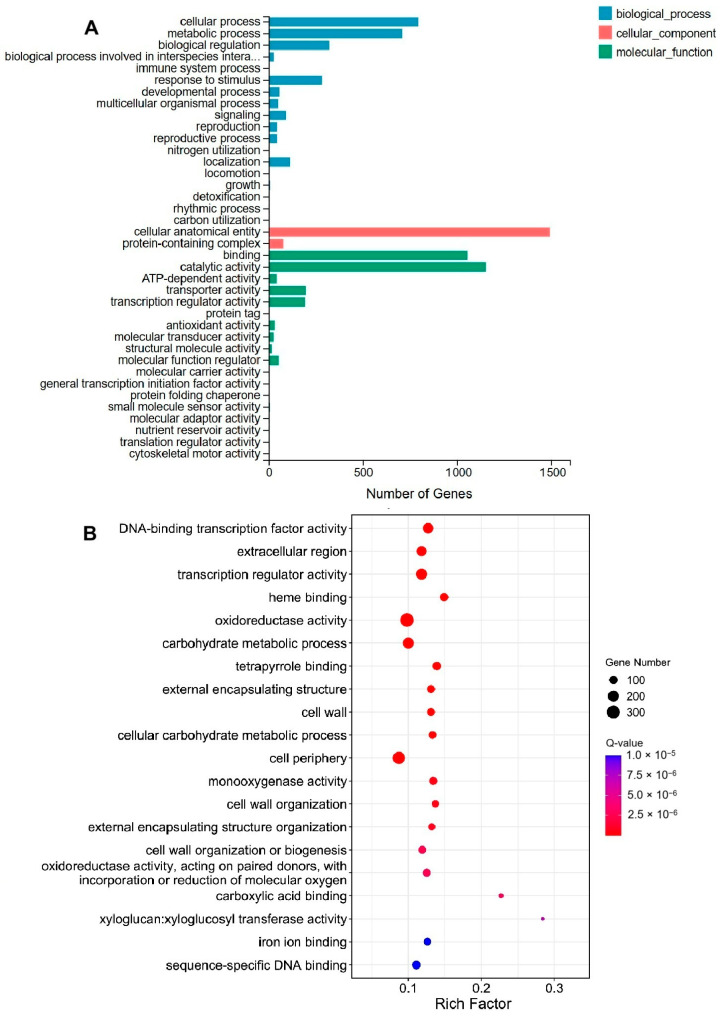
GO analysis of DEGs. (**A**) In the GO classification diagram, the *X*-axis represents the number of genes annotated to the GO term, and the *Y*-axis represents GO functional categories. (**B**) GO enrichment analysis, with the *X*-axis representing the enrichment ratio, the *Y*-axis representing GO terms, the size of the bubbles indicating the number of DEGs annotated to a particular GO term, and the color representing the enrichment Q-value, where the greener the color, the smaller the Q-value. The omitted content in Figure 4A is “biological process involved in interspecies interaction between organisms”.

**Figure 5 ijms-26-05360-f005:**
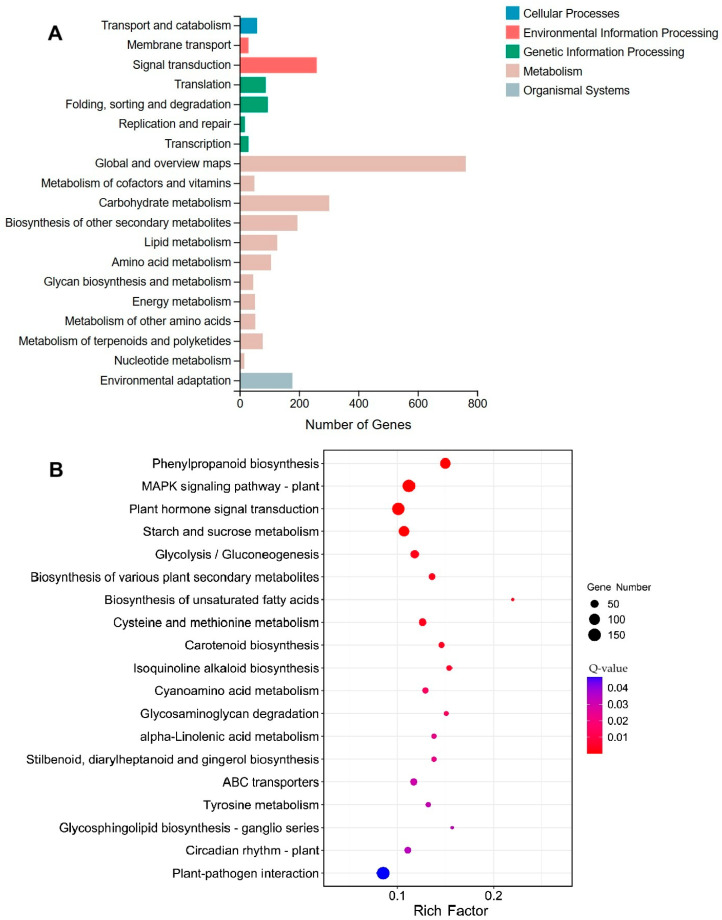
KEGG pathway analysis of DEGs. (**A**) KEGG pathway classification diagram; the *X*-axis represents the number of genes annotated to the KEGG pathway category and the *Y*-axis represents the KEGG pathway category. (**B**) KEGG pathway enrichment analysis; the *X*-axis represents the enrichment ratio, the *Y*-axis represents the KEGG pathway, the size of the bubbles represents the number of genes annotated to the KEGG pathway, and the color represents the enrichment Q-value, with darker colors indicating smaller Q-values.

**Figure 6 ijms-26-05360-f006:**
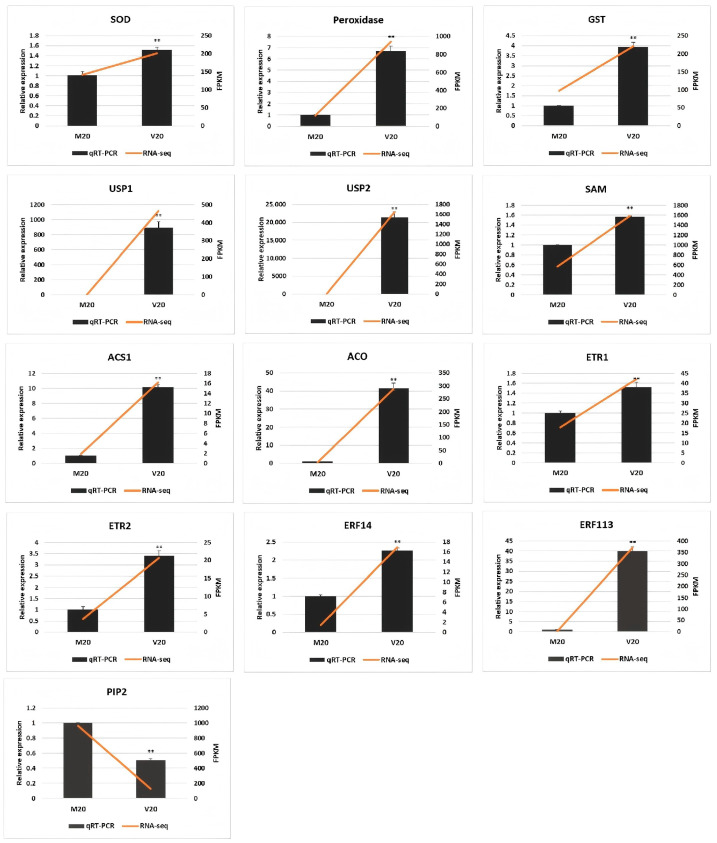
Validation of RNA-seq data by qRT-PCR. M20: normal callus; V20: hyperhydric callus; SOD: superoxide dismutase; GST: glutathione S-transferase; USP: universal stress protein; SAM: S-adenosylmethionine synthetase; ACO: 1-aminocyclopropane-1-carboxylate (ACC) synthase; ETR: ethylene receptor; ERF: ethylene-responsive factor; PIP2: aquaporin PIP2. Error bars indicate the standard error (*n* = 3), ** denotes a very significant difference (*p* < 0.01).

**Figure 7 ijms-26-05360-f007:**
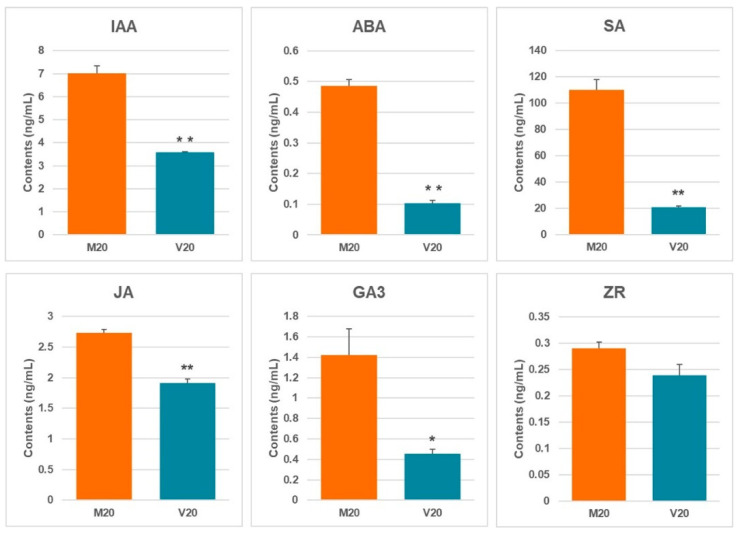
Analysis of endogenous hormone content: auxin (IAA), abscisic acid (ABA), salicylic acid (SA), jasmonic acid (JA), gibberellin (GA3), and zeatin (ZR). * denotes a significant difference (*p* < 0.05), ** denoted an extremely significant difference (*p* < 0.01).

**Table 1 ijms-26-05360-t001:** Summary of sequencing data.

Sample	Raw Reads (M)	Clean Reads (M)	Clean Bases (Gb)	Q20 (%)	Q30 (%)	Mapping Ratio (%)
M20a	47.19	43.32	6.5	97.9	92.92	96.35
M20b	43.69	40.7	6.11	98.47	94.33	96.51
M20c	45.44	42.55	6.38	97.83	92.61	96.76
V20a	42.7	40.12	6.02	98.38	94.02	96.72
V20b	47.19	43.3	6.5	98.21	93.59	96.43
V20c	48.93	44.2	6.63	98.02	93.21	96.08

**Table 2 ijms-26-05360-t002:** Differentially expressed genes in sample statistics.

Group	Total	Upregulated Genes	Downregulated Genes
M20 vs. V20	2698	1667	1031

## Data Availability

The original contributions presented in this study are included in the article/Appendix A. Further inquiries can be directed to the corresponding author(s).

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
