# Peer review of "Transcriptome and Endogenous Hormone Analysis Reveals the Molecular Mechanism of Callus Hyperhydricity in Flax (Linum usitatissimum L.)"

_ijms, 2025, doi:10.3390/ijms26115360_

Round 1

Reviewer 1 Report

Comments and Suggestions for Authors

This study analyzes the molecular mechanisms behind hyperhydricity in flax (Linum usitatissimum L.) calli. It uses transcriptomic analysis and endogenous hormone measurements to identify 2,698 differentially expressed genes between normal and hyperhydric calli in Linum usitatissimu. The work is interesting; I find it very well structured; the objectives are consistent with the results and support the conclusion.

The topic is relevant and it is interesting because there are no studies about the genes responsible for the formation of hyperhydricity, their expression levels, or their relationships with the expression of different hormones related to Linum's normal development and when hyperhydricity occurs.

Currently, there is no information on the genes responsible for the formation of hyperhydricity in Linum and in general, little is known about the genetic interactions that promote the development of these deformations so the research is important.

If compared with other published material, similar information involving different gene families, their expression levels, and the hormones involved had not been reported; so far, for Linum, only a few gene families had been identified.

I did not identify any conflicts or errors in the methodological part, so I made no observations regarding this point.

The results meet the objective of the work, are adequately argued and discussed based on other works, its objective was clear, to identify and compare gene families between two different developments of Linum, and they demonstrate this with the results.

The references are appropriate.

There are only a few details that need to be corrected to improve the manuscript:

Agrobacterium in italics, line 61.

Although the full name is in the subtopic title, you can add the abbreviation immediately (DEGs) to make it clear that the initials correspond to Differentially Expressed Genes (lines 92 and 93).

Correct Figure 4B so that the X-axis scale starts at least 0.05 from 0 so that the green circle is not below the "cell periphery" on the Y-axis.

Similarly, for Figure 5B, start at 0 or 0.05 so that the blue circle is not below "Plant-pathogen interaction."

Add the full name: Indole-3-acetic acid, followed by the abbreviation IAA (line 169).

Make Figure 6 larger so the graphics can be seen in better detail.

Arabidopsis thaliana in italics (line 246)

Author Response

1.Summary

Thank you very much for taking the time to review this manuscript. Please find the detailed responses below and the corresponding revisions highlighted in the attachment.

2. Point-by-point response to Comments and Suggestions for Authors

Comments 1: Agrobacterium in italics, line 61.

Response 1: Thank you for pointing this out. We have corrected Agrobacterium to italics in the article. For details, see line 62.

Comments 2: Although the full name is in the subtopic title, you can add the abbreviation immediately (DEGs) to make it clear that the initials correspond to Differentially Expressed Genes (lines 92 and 93).

Response 2: Thank you for your advice. We have added the abbreviation for Differentially Expressed Genes in the article. Please refer to line 93 for details.

Comments 3: Correct Figure 4B so that the X-axis scale starts at least 0.05 from 0 so that the green circle is not below the "cell periphery" on the Y-axis.

Response 3: Thank you for pointing this out. We have made revisions to Figure 4B, as detailed in line 134.

Comments 4: Similarly, for Figure 5B, start at 0 or 0.05 so that the blue circle is not below "Plant-pathogen interaction".

Response 4: Thank you for pointing this out.We have made revisions to Figure 5B, as detailed in line 141.

Comments 5: Add the full name: Indole-3-acetic acid, followed by the abbreviation IAA (line 169).

Response 5: Thank you very much. The full name of IAA, Indole-3-acetic acid, has been added. For details, please refer to line 171.

Comments 6: Make Figure 6 larger so the graphics can be seen in better detail.

Response 6: Thank you very much for your suggestion. Figure 6 has been enlarged. For details, please refer to line 157.

Comments 7: Arabidopsis thaliana in italics (line 246)

Response 7: Thank you for pointing this out. We have corrected Arabidopsis thaliana to italics in the article. For details, see line 268.

Reviewer 2 Report

Comments and Suggestions for Authors

This is the first comprehensive molecular analysis of hyperhydricity in flax, addressing a significant gap in the literature. The combination of RNA-seq, qRT-PCR validation, and hormone quantification provides a multi-faceted approach to understanding the phenomenon. Excellent sequencing metrics (Q20 >97%, Q30 >92%, mapping rate 96.47%) demonstrate technical rigor. qRT-PCR validation of 13 genes with 12/13 showing consistent resultsconfirms transcriptomic reliability. Well-documented phenotypic differences between normaland hyperhydric callus. However, several areasrequire improvement before publication.

The paper lacks sufficient detail about mRNA and hormone extraction methodology. Critical parameters such as RNA extraction kit producer, hormone extraction efficiency, recovery rates, and method validationare missing.

Culture condition in terms of insufficient detail about environmental conditions that may influence hyperhydricity development.

The downregulation of PIP2 aquaporin contradicts some previous studies,but the discussion doesn't adequately address these discrepancies beyond temporal considerations.

Limited correlation analysis between hormone levels andexpression of genes in corresponding biosynthetic pathways.

Author Response

1.Summary

Thank you very much for taking the time to review this manuscript. Please find the detailed responses below and the corresponding revisions highlighted in the attachment.

2. Point-by-point response to Comments and Suggestions for Authors

Comments 1: The paper lacks sufficient detail about mRNA and hormone extraction methodology. Critical parameters such as RNA extraction kit producer, hormone extraction efficiency, recovery rates, and method validationare missing.

Response 1: Thank you for pointing this out. The manufacturer of the RNA extraction kit and the details of the hormone extraction method have been added to the article. For details, see line 293 and lines 326-338.

Comments 2: Culture condition in terms of insufficient detail about environmental conditions that may influence hyperhydricity development.

Response 2: Thank you for your advice. The culture conditions for callus have been added to the article. For details, please refer to lines 286-288.

Comments 3: The downregulation of PIP2 aquaporin contradicts some previous studies,but the discussion doesn't adequately address these discrepancies beyond temporal considerations.

Response 3: Thank you very much for your suggestion. The explanation for the contradictory research findings has been added to the discussion section. For details, please refer to lines 247-261.

Comments 4: Limited correlation analysis between hormone levels andexpression of genes in corresponding biosynthetic pathways.

Response 4: Thank you for pointing this out. We have added the correlation analysis between hormone levels and gene expression in the corresponding biosynthetic pathways to the discussion. For details, please refer to lines 230-238.

Reviewer 3 Report

Comments and Suggestions for Authors

It's a very interesting topic. The material is clear and easy to follow. A lay reader can easily understand it. The diagrams and tables are good and easy to understand. The methods used are good. The evaluation of the findings is good. 

Author Response

I sincerely appreciate you taking the time to review the article amidst your busy schedule and providing valuable affirmation and encouragement! Your recognition is a tremendous source of motivation for me, and I will continue to maintain a rigorous attitude and strive to improve the quality of my research. Thank you once again for your support!